# Development and Validation of a Clinically Relevant Workflow for MR-Guided Volumetric Arc Therapy in a Rabbit Model of Head and Neck Cancer

**DOI:** 10.3390/cancers12030572

**Published:** 2020-03-01

**Authors:** Eftekhar Rajab Bolookat, Harish Malhotra, Laurie J. Rich, Sandra Sexton, Leslie Curtin, Joseph A. Spernyak, Anurag K. Singh, Mukund Seshadri

**Affiliations:** 1Laboratory for Translational Imaging, Center for Oral Oncology, Roswell Park Comprehensive Cancer Center, Buffalo, NY 14263, USA; Eftekhar.RajabBolookat@roswellpark.org (E.R.B.); Laurie.Rich@pennmedicine.upenn.edu (L.J.R.); 2Department of Radiology—Medical Physics Program, University at Buffalo—Jacobs School of Medicine and Biomedical Sciences, 955 Main Street, Buffalo, NY 14203, USA; Harish.Malhotra@roswellpark.org (H.M.); Joseph.Spernyak@roswellpark.org (J.A.S.); 3Department of Radiation Medicine, Roswell Park Comprehensive Cancer Center, Buffalo, NY 14263, USA; Anurag.Singh@roswellpark.org; 4Laboratory Animal Shared Resource, Roswell Park Comprehensive Cancer Center, Buffalo, NY 14263, USA; Sandra.Sexton@roswellpark.org (S.S.); Leslie.Curtin@roswellpark.org (L.C.); 5Department of Cell Stress Biology, Roswell Park Comprehensive Cancer Center, Buffalo, NY 14263, USA; 6Department of Dentistry and Maxillofacial Prosthetics, Roswell Park Comprehensive Cancer Center, Buffalo, NY 14263, USA

**Keywords:** image-guided radiation therapy, VMAT, MRI, HNSCC

## Abstract

There is increased interest in the use of magnetic resonance imaging (MRI) for guiding radiation therapy (RT) in the clinical setting. In this regard, preclinical studies can play an important role in understanding the added value of MRI in RT planning. In the present study, we developed and validated a clinically relevant integrated workflow for MRI-guided volumetric arc therapy (VMAT) in a VX2 rabbit neck tumor model of HNSCC. In addition to demonstrating safety and feasibility, we examined the therapeutic impact of MR-guided VMAT using a single high dose to obtain proof-of-concept and compared the response to conventional 2D-RT. Contrast-enhanced MRI (CE-MRI) provided excellent soft tissue contrast for accurate tumor segmentation for VMAT. Notably, MRI-guided RT enabled improved tumor targeting ability and minimal dose to organs at risk (OAR) compared to 2D-RT, which resulted in notable morbidity within a few weeks of RT. Our results highlight the value of integrating MRI into the workflow for VMAT for improved delineation of tumor anatomy and optimal treatment planning. The model combined with the multimodal imaging approach can serve as a valuable platform for the conduct of preclinical RT trials.

## 1. Introduction

Radiation therapy (RT) remains an important component of the standard of care for patients with head and neck squamous cell carcinomas (HNSCC). Clinical studies have shown that volumetric modulated arc therapy (VMAT) can ensure precise radiation delivery to tumors with reduced treatment times while sparing organs at risk (OAR) [1,2,3,4]. In VMAT, the radiation beam is continuously reshaped with changing intensity as it moves around the body to ensure highly conformal dose distribution to the target tissue while sparing surrounding tissues. While computed tomography (CT) has been traditionally used for radiation treatment planning, there has been increased interest in the use of magnetic resonance imaging (MRI) for guiding RT in the clinical setting [5,6,7]. In this regard, preclinical studies can play an important role in understanding the added value of MRI for radiation treatment planning and response assessment. Such studies could also assist in the optimization and integration of MRI into the clinical workflow [8,9]

Unfortunately, the vast majority of radiation studies conducted in rodent models do not mimic clinical practice. Experimental studies typically employ crude radiation delivery methods such as orthovoltage X-rays or ^137^Cesium irradiators [10,11]. These two-dimensional RT (2D-RT) methods lack 3D image guidance for treatment planning and therefore do not allow for precise dose calculations to target volumes or OAR [12,13]. This is particularly critical in HNSCC given the abundance of critical structures (e.g., carotid artery) in the head and neck region [14]. The use of 2D-RT methods is not standard for humans but is widely used for preclinical studies in small and large animal models. As a result, the clinical relevance or translational insight gained from these preclinical RT studies remains limited [8,9,15]. To address this limitation, image-guided micro irradiators have been recently developed. These commercially available RT platforms enable precision radiotherapy through built-in image guidance for treatment planning [16]. However, these CT-guided radiation research platforms are limited to small animals such as mice and rats [17,18,19].

While rodent tumor models are cost-effective and widely utilized as preclinical tools to evaluate therapeutic efficacy, their value in predicting clinical activity of novel agents remains controversial. Furthermore, important physiological differences exist between mice and humans that limit the extrapolation of findings from preclinical studies to clinical trials, particularly for RT studies [15]. For example, murine tumors are typically a few millimeters in size compared to human HNSCC that are typically several centimeters in thickness. Similarly, in the head and neck region, differences in salivary gland anatomy between rodents and humans exist that should be considered in the context of RT treatment planning and toxicity assessment [20]. Given these limitations of small animal models, there has been increased interest in the utilization of large animal models of HNSCC for preclinical studies [21,22,23]. In this regard, we have recently reported on a novel orthotopic rabbit model of HNSCC based on surgical transplantation of Shope cottontail rabbit papilloma virus associated VX2 tumors into the neck of rabbits [24]. Although the VX2 rabbit model has been previously utilized for imaging and therapeutic response studies [25,26,27], the use of this model for image-guided SBRT or VMAT of head and neck tumors has not been reported.

The overall goal of the present study was to develop and validate a clinically relevant integrated workflow for combined MRI and CT-guided VMAT in the VX2 neck tumor model of HNSCC. In addition to demonstrating safety and feasibility, we examined the therapeutic impact of MR-guided VMAT compared to conventional 2D-RT.

## 2. Results

### 2.1. MR-CT and US Imaging of VX2 Neck Tumors—Study Design and Workflow

The study design and sequence of imaging examinations for the study is shown in Figure 1. Clinical examination provided visual confirmation of tumor growth in all the animals over the three-week period post implantation. Baseline contrast-enhanced MRI (CE-MRI) and non-contrast enhanced CT exams were performed on Day 0 for treatment planning and the developed treatment plan for VMAT was delivered on Day 1. Change in tumor volume evaluated using B-mode US images acquired at baseline, one, two, four, and eight weeks post RT. In addition to radiation delivery, treatment responses and toxicity with IG-VMAT were compared to conventional two-dimensional RT (2D-RT). 

### 2.2. MR-Guided VMAT of VX2 Tumors

The workflow for MR-guided VMAT of orthotopic VX2 neck tumors is shown in Figure 2. Pretreatment MR images were acquired three weeks post implantation and included T1-weighted acquisitions before and after administration of Gd-DTPA (Magnevist^®^, gadopentetate dimeglumine, Bayer, Whippany, NJ, USA). Post contrast scans showed predominant enhancement in the tumor periphery allowing for clear delineation of tumor boundary (Figure 2D). Following completion of MRI acquisition, animals were allowed to recover for a few hours and subsequently, a treatment planning CT scan was performed (Figure 2A,B and Appendix A). MR images were rigidly co-registered with CT using anatomic landmarks as a guide to ensure prompt treatment planning and radiation delivery within a 24-h period. As evident from the individual images, CE-MRI provided excellent soft tissue contrast to allow for tumor delineation. The tumor was traced on the contrast-enhanced T1-weighted MR images and manually co-registered to the treatment planning CT image using anatomic landmarks as a guide (Appendix A). The segmentation was confirmed on the fused image. Animals in the 2D RT group did not undergo any imaging examination.

### 2.3. Treatment Planning and Dosimetry

The planned target volume (PTV) was defined as 0.5 cm expansion of the gross tumor volume (GTV). Planning objectives were set to deliver single fraction of 30 Gy to cover at least 95% of PTV. Organs at risk (OARs) were defined as the skin, trachea, parotid glands, brain, and spinal cord. Dose constraints were set according to Radiation Therapy Oncology Group (RTOG) protocols, which were defined based on single fraction SRS. To check if planning objectives were met, actual dose distributions in PTV and OAR were evaluated by dose volume histograms and 3D isodose lines. If the optimization result was not acceptable, optimization was rerun with modified parameters for priority and dose constraints.

During RT, the rabbits were positioned in relation to the planning CT scan and it was confirmed by use of an on-board cone beam CT (CBCT) using the planning CT set as a reference CT (Figure 3). The CBCT was overlaid on the reference CT and necessary corrections to bring the rabbit in desired treatment position by moving the treatment couch. Deviation from the planning CT was set to 2.0 mm as the tolerance level to reposition and reimage the rabbits. The developed treatment plans for rabbits undergoing MR-guided VMAT fulfilled the requested objective for the target coverage and dose constraints to OAR (Appendix A). As can be seen from the dose distribution and the dose volume histogram (DVH) in Figure 4, the MR-based treatment plan result in homogenous dose delivery to the GTV and reduced dose to tissues outside the PTV. Comparative evaluation of DVH of CT- and MR-based treatment plans showed that tracheal volume receiving 18 Gy (dose limit) increased from 0% in the MR-based plan to 85% in the CT-based plan (Appendix A).

At least 95% of PTV was covered by 30 Gy in all three rabbits (Rb1, Rb2, and Rb3) treated using VMAT. The volume outside PTV that received 105% of prescribed dose (31.5 Gy) was zero for all rabbits. Max dose beyond PTV (PTV + 2 cm) was 55–56% of the prescribed dose, which met the protocol limit. Table 1 shows dose constraints for PTV and the dose to OAR obtained by DVH analysis is presented in Table 2. The dose received by OARs was within their tolerance limit.

### 2.4. Treatment Outcome and Toxicity—MR-Guided VMAT vs. 2D RT

To examine the therapeutic impact of MR-guided VMAT, longitudinal US was performed over an eight-week observation period. B-mode US was performed and tumor volume measurements were obtained for animals in control, MR-guided VMAT, and 2D-RT groups (Figure 5).

Good concordance was seen between tumor volume measurements obtained with US and MRI (Appendix A). Figure 5A shows longitudinal US images of a single animal in each group over the eight-week period. Corresponding tumor volume curves are shown in Figure 5B. Control animals showed a rapid increase in tumor volume within 10 days that was associated with considerable weight loss necessitating euthanasia. Tumor bearing rabbits treated with 2D RT using the orthovoltage radiation system showed growth inhibition. However, animals had to be euthanized at two weeks post RT due considerable morbidity (weight loss, poor appetite, and radiation dermatitis). In comparison, animals that underwent VMAT showed an initial tumor growth inhibition (one week post RT), which was followed by a marked reduction in tumor volume (~85–100% reduction at eight weeks post RT).

## 3. Discussion

The overall goal of radiation therapy is to safely and accurately deliver appropriate radiation dose to the tumor while minimizing the dose to surrounding normal tissues. In the present study, we established a preclinical protocol that closely mimics clinical workflow for combined MR and CT to enable accurate tumor delineation and dosimetry in a VX2 rabbit neck tumor model of HNSCC. Our results demonstrate the feasibility and therapeutic impact of IG-VMAT compare to 2D RT. The model combined with the multimodal imaging approach can serve as a valuable platform for the conduct of preclinical RT trials.

The field of radiation medicine has experienced tremendous advancements over the last two decades including the development of state-of-the-art CT scanners for 3D imaging and powerful treatment planning systems [29]. Improved radiation delivery techniques such as IMRT and VMAT with image guidance are being increasingly evaluated in the clinical setting for patients with solid tumors including HNSCC [2,3,4,30,31,32]. Unfortunately, image-guided RT delivery methods are not routinely employed in preclinical studies. Studies in rodent tumor models have typically utilized orthovoltage systems that are susceptible to dosimetry errors [12,13]. Commercial CT-guided RT platforms have been recently developed and are being increasingly utilized for studies in small animal models of cancer [16,17,18,19]. While mouse models are widely utilized in research, the clinical or translational value of radiation studies in murine systems remains unclear. As a result, there has been increased interest in developing and applying canine (dogs), feline (cats), and leporine (rabbits) models of cancer for preclinical research [21,22,23,24,25,26,27]. However, the literature on the implementation of IG-VMAT in large animal models of cancer, especially HNSCC, is limited.

Since commercially available image-guided RT systems are not amenable to conducting studies using large animals, we developed and refined a protocol based on MR-guided VMAT in rabbits bearing VX2 tumors in the neck. The excellent soft tissue contrast afforded by MRI allowed for precise tumor delineation while the linear accelerator with onboard image guidance enabled precise delivery to the target volume. Despite the close association of the tumor to critical structures in the neck such as the trachea, spinal cord, and brain, a single high dose of 30 Gy was safely delivered under MR-guidance and resulted in a marked reduction in tumor volume. In recent years, highly conformal VMAT can be potentially delivered using a hypofractionated regimen for patients with recurrent disease following fractionated radiation [33]. We therefore evaluated our MR-guided RT approach using a single high dose to obtain proof-of-concept. While MR-guided RT has been reported in rat models of prostate, pancreas, and brain tumors [34,35,36], to the best of our knowledge, the potential of this approach has not been previously demonstrated in HNSCC. Our results also highlight the superior targeting ability of VMAT with minimal dose to OAR. In contrast, 2D-RT did not allow adequate sparing of these tissues and resulted in notable morbidity within a few weeks of RT. Consistent with our observations and these clinical studies, Wang et al. reported the potential of VMAT and simultaneous integrated boost technique is feasible in the rabbit VX2 limb tumor model [37]. Similarly, Dolera et al. reported on the safety of VMAT for treating rabbit thymomas as well as brain tumors in pets [38].

Our study limitations warrant recognition. Since our main goal was to establish and validate the feasibility of a clinically-relevant workflow, the sample size for our cohorts was relatively small. While we were able to achieve good co-registration to enable MR-guided treatment planning, we did not investigate the potential geometric distortions among the MR, CT and CBCT images. Future studies should evaluate and correct for these potential distortions to further improve the accuracy of MR-guided RT [35]. Second, our study was focused on evaluation of primary tumor response to RT. It would be clinically insightful to examine the long-term survival along with evaluation of disease recurrence or metastases with MR-guided VMAT. As such, integrating MR into the clinical RT workflow is being actively investigated in the clinical setting. Our platform could also be valuable in the conduct of preclinical trials to evaluate and optimize the combination of RT with targeted agents and immunotherapy.

## 4. Materials and Methods 

### 4.1. Animals

Adult male, specific-pathogen free, New Zealand White (NZW) rabbits (body weight range, 1.4–1.8 kg) were purchased from Charles River Corporation (Saint Constant, Quebec, Canada). Animals were acclimatized for one week before experiment. Animals were housed in individual stainless-steel cages containing noncontact bedding (24” w × 24” d × 16” h) under 12-h light and dark cycles. Rabbits were fed high-fiber diet (Rabbit diet # 2031; Harlan Teklad, KY, USA) and purified water and dietary enrichment consisted of Timothy Hay cubes and kale. Environmental enrichment in the form of manipulanda (Bio-Serv, Flemington, NY, USA) was also provided.

### 4.2. Surgical Procedure for Establishing VX2 Tumors in the Neck

We recently described the surgical procedure for establishing VX2 tumors in the neck of rabbits [24]. Prior to the surgical procedure, topical lidocaine and prilocaine anesthetic cream (Hi Tech Pharmacal Co. Inc., Amityville, NY, USA) was applied to the ear. Animals were sedated using acepromazine maleate (0.3 mg/kg, i.v. Phoenix Pharm Inc., Burlingame, CA, USA). Induction and maintenance of anesthesia were performed using inhalational isoflurane (Patterson Logistics Services Inc., Mount Joy, PA, USA). Preoperative analgesia was achieved by intramuscular administration of buprenorphine hydrochloride (0.05 mg/kg; Patterson Logistics Services Inc., Mount Joy, PA, USA). An incision was made on the skin along the length of the neck followed by blunt dissection of subcutaneous tissue and fascia. A piece of sterile VX2 tumor tissue (approximately 2 mm × 2 mm) from a donor rabbit was placed within a pocket created within the lateral aspect of the sternohyoid muscle tissue under anesthesia. The pocket was closed with a 6-0 Vicryl suture and the skin wound closed with tissue glue and ethilon suture. All surgical procedures were performed under aseptic conditions in an AAALAC-accredited facility and in accordance with protocols approved by the Institutional Animal Care and Use Committee (Rb 1285; Approval date: 29 January 2015). Following surgical implantation, animals were monitored daily for general health, signs of distress, and/or tumor growth. Humane end points for euthanasia included tumor size greater than 4 cm in diameter and/or any signs of distress.

### 4.3. Magnetic Resonance Imaging

MR imaging examinations were performed using a 4.7T/33 cm horizontal bore magnet (GE NMR Instruments, Fremont, CA, USA) incorporating AVANCE digital electronics (Bruker Biospec with Paravision 3.0.2; Bruker Biospin., Billerica, MA, USA). Animals were sedated using acepromazine maleate followed by inhalational anesthesia using Isoflurane (Benson Medical Industries, Markham, ON, Canada). A catheter was placed into the marginal auricular vein for administration of contrast. The rabbits were secured in a form-fitted, MR compatible cradle in the supine position along with a phantom (1% agarose with NMR tube). T1-weighted, 3D spoiled gradient recalled echo (SPGR) scans were performed with the following parameters: TR = 15 ms; TE = 3 ms; flip angle = 25°; field of view = 16 cm × 12 cm × 12 cm; matrix size = 192 × 128 × 128; number of averages (NEX) = 2; and bandwidth = 83.3 kHz. Images were obtained before and up to 8 min post-gadolinium injection (Magnevist: 0.05 mmol/kg, flush 10 mL saline after injection administered intravenously). Following MR image acquisition, raw image sets were transferred to a processing workstation and converted into Analyze™ format (Analyze 7.0; AnalyzeDirect, Overland Park, KS, USA). Image segmentation included a region of interest traced around the entire tumor area excluding the surrounding skin.

### 4.4. Radiation Therapy

Ten rabbits were used for experimental studies. Three weeks post implantation, rabbits were assigned to control (*n* = 4), MR-guided VMAT (*n* = 3), or 2D-RT (*n* = 3) groups. For VMAT, rabbits were anesthetized (isoflurane, 2%) and placed in a supine position in a Big Bore CT scanner (GE discovery RT 16 slices, slice thickness: 0.125 cm, 120 kV, 99 mA, FOV 35 cm, matrix size 512 × 512). CT images were imported into the Eclipse treatment planning system (version 13.6, Varian Medical Systems, Palo Alto, CA, USA). MRI and CT scans were fused manually using anatomic landmarks as a guide through the image registration algorithm provided in the treatment planning system. Two full arcs were designed in clockwise rotation (181° to 179°) and in counter-clockwise rotation (179° to 181°). The collimator angles were 30° and 330° for the first and second arc, respectively. Anisotropic Analytical Algorithm (AAA) with a calculation grid set to 2.5 mm was used for dose calculation with heterogeneity correction. RT was applied with 6 MV photon beams and maximum dose rate of 600 MU/min using Varian linear accelerator (Triology, Varian Medical Systems, Palo Alto, CA, USA) with the Millennium 120-mutileaf collimator. Conventional RT of the prescribed dose (30 Gy calculated by a single point along the central axis) was delivered by two opposite lateral fields using the Philips RT 250 Orthovoltage X-ray unit (Philips Medical Systems, Andover, MA, USA) equipped with an aluminum filter and operating at 250 kV/17.7 mA. The dose rate at this setting was ~0.68 Gy/min. Animals were monitored with a closed-circuit TV during irradiation and observed after completion of treatment to ensure full recovery.

### 4.5. Ultrasound

Experimental US examination was performed using a commercially available high-frequency ultrasound system (FujiFilm VisualSonics Inc., Toronto, Canada) consisting of a 15 MHz transducer synchronized to a micro-ultrasound system and a work station to process and reconstruct the images. Animals were positioned on their back for imaging and three-dimensional (3D) B-mode US images were acquired to calculate tumor volume. Animal vitals were monitored continuously during the imaging procedure and following completion of imaging to ensure full recovery.

## 5. Conclusions

We demonstrated the ability of MRI-guided RT in a large animal model of HNSCC. Our results highlight the value of integrating MRI into the workflow for VMAT for improved delineation of tumor anatomy and optimal treatment planning. Our clinically relevant model system and workflow could enable future evaluation of adaptive planning based on functional MRI methods such as blood oxygenation level dependent MRI (BOLD-MRI) and tissue oxygenation level dependent MRI (TOLD-MRI).

## Figures and Tables

**Figure 1 cancers-12-00572-f001:**
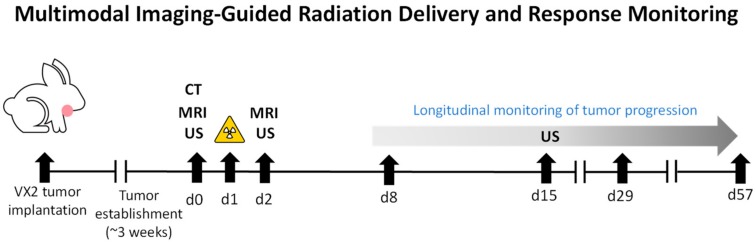
Multimodal imaging-guided radiation delivery and response monitoring. Schematic shows the study design and sequence of imaging examinations conducted in VX2 tumor bearing rabbits. Combined magnetic resonance imaging (MRI) and computed tomography (CT) were used for radiation treatment planning while ultrasound (US) was used to assess changes in tumor growth. Radiation therapy (indicated by the yellow symbol) was delivered 24 h following baseline line image acquisition.

**Figure 2 cancers-12-00572-f002:**
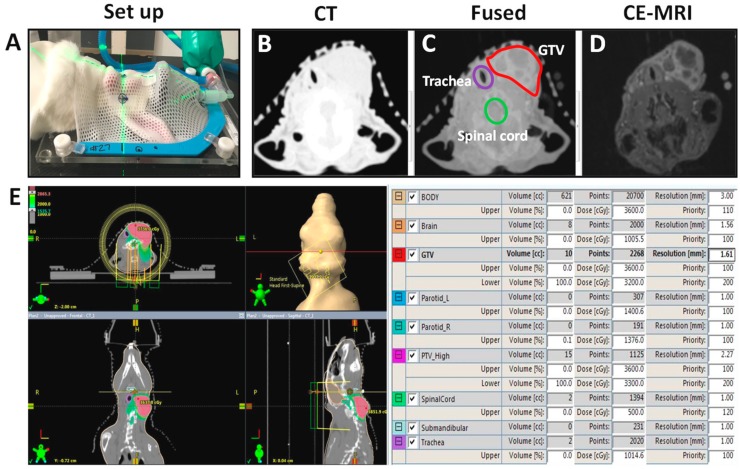
Setup and workflow for MRI-guided VMAT. (**A**) Setup for CT stimulation of VX2 tumor bearing rabbits. Anesthetized rabbits were positioned on the CT table and a thermoplastic face mask was molded to immobilize the tumor, as shown in the photograph. Axial CT slice (**B**) and corresponding T1-weighted CE-MRI (**D**) examinations were performed to enable image-guided VMAT. Images were spatially co-registered (fused) (**C**) to enable accurate delineation of target volume. Gross tumor volume (GTV) (red) along with OAR including trachea (purple) and spinal cord (green) were traced as shown in the fused image to develop and optimize radiation treatment plans (**E**). Parameters for plan optimization included upper and lower dose limits for each organ and calculation of dose distribution.

**Figure 3 cancers-12-00572-f003:**
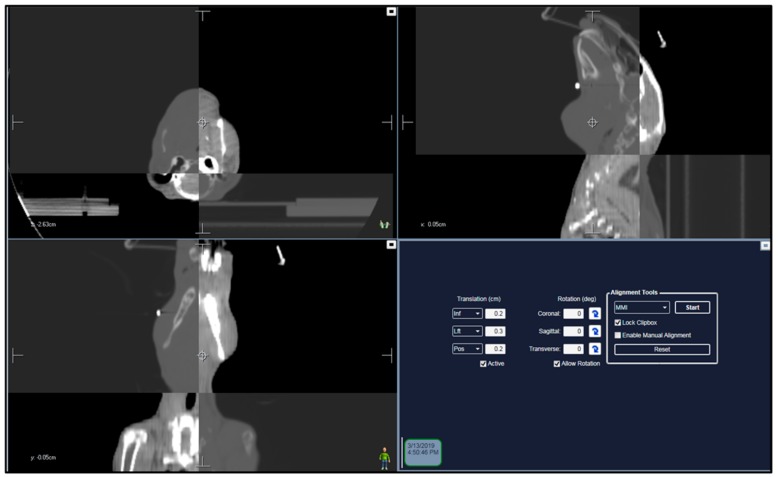
Co-registration of CBCT and treatment planning CT. Panel of images show axial, sagittal, and coronal sections of cone beam CT images aligned with treatment planning CT for verification prior to radiation delivery. Any shifts in position were calculated and applied onto the treatment couch to ensure accurate targeting of the tumor.

**Figure 4 cancers-12-00572-f004:**
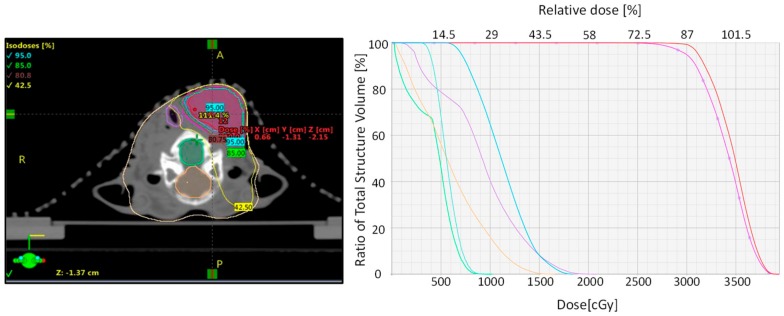
Dose distribution map and dose volume histogram of VX2 tumors. Dose distribution map for tumor volume and OAR displayed as isodoses (%) is shown on the left. Corresponding dose volume histogram (DVH) for radiation dose to PTV (pink), GTV (red), right and left parotid glands (blue), spinal cord (green), brain (orange), and trachea (purple) is shown on the right.

**Figure 5 cancers-12-00572-f005:**
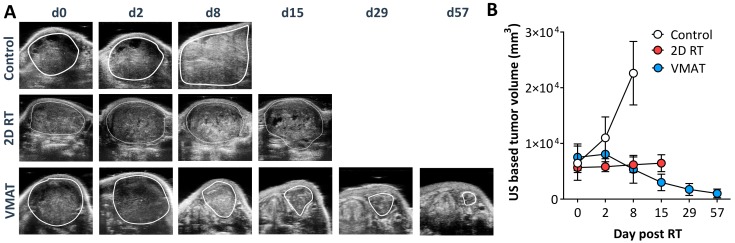
US-based monitoring of VX2 response to MR-guided VMAT. (**A**) Longitudinal B-mode US images of a representative animal from control, VMAT, and 2D RT groups. Tumor is outlined in white on all the images. Columns represent days post RT (d, days) (**B**) Corresponding tumor volume measurements at different time points post RT in VMAT, 2D-RT, and control cohorts. Animals treated by VMAT showed 85–100% reduction in tumor volume by Day 57. However, animals in 2D-RT cohort demonstrated steady increase in tumor volume and they were euthanized 14 days post RT due to significant morbidity.

**Table 1 cancers-12-00572-t001:** Planning target volume dosimetric parameters generated through DVH analysis (from NRG-BR002 Table 5–5; ClinicalTrials.gov NCT02364557 [28]).

Prescription Dose Constraints								
PTV (cc)	Rb1	8				Rx IDL (%):	Rb1	80
	Rb2	10					Rb2	80
	Rb3	19.3					Rb3	90
100% Rx Dose			50% Rx Dose			105% Rx Dose outside PTV		
30 Gy vol (cc)	Rb1	7.5	15Gy vol (cc)	Rb1	8	31.5 Gy vol (cc)	Rb1	0
	Rb2	9.5		Rb2	10		Rb2	0
	Rb3	18.4		Rb3	19.3		Rb3	0
Rx Isodose Coverage								
Parameter			Limit		Actual		Protocol Limit Met?	
Rx Isodose (%)			≥ 60%, ≤ 90%		Rb1	80	YES	
					Rb2	80	YES	
					Rb3	90	YES	
% PTV covered by 30 Gy			≥ 95		Rb1	95	YES	
					Rb2	95	YES	
					Rb3	95	YES	
Vol outside PTV ≥ 105% Rx dose (cc)			0		Rb1	0	YES	
					Rb2	0	YES	
					Rb3	0	YES	
Conformality - Vol rx IDL / Vol PTV			≤ 1.2		Rb1	0.95	YES	
					Rb2	0.95	YES	
					Rb3	0.96	YES	
Max dose beyond PTV + 2.0 cm (% Rx Dose)			Rb1	≤ 58.52	56.6		YES	
			Rb2	≤ 60.24	56.0		YES	
			Rb3	≤ 68.26	53.3		YES	
Vol 50% Rx Dose / Vol PTV			Rb1	≤ 5.77	1		YES	
			Rb2	≤ 5.67	1		YES	
			Rb3	≤ 5.18	1		YES	

**Table 2 cancers-12-00572-t002:** Dose Limits on Critical Structures (from NRG-BR002 and RTOG).

Dose Constraints for OAR.					
Organ	Parameter	Limit	Actual		Protocol Limit Met?
**Spinal Cord**	Volume > 8.0 Gy	≤ 1.20 cc	Rb1	0.00 cc	Yes
			Rb2	0.00 cc	Yes
			Rb3	0.00 cc	Yes
	Volume > 10.0 Gy	≤ 0.35 cc	Rb1	0.00 cc	Yes
			Rb2	0.00 cc	Yes
			Rb3	0.00 cc	Yes
**Trachea**	Volume > 17.4 Gy	≤ 4.00 cc	Rb1	1.00 cc	Yes
			Rb2	0.00 cc	Yes
			Rb3	0.00 cc	Yes
**Skin (surface − 2 mm)**	Volume > 25.5 Gy	≤ 10.00 cc	Rb1	7.69 cc	Yes
			Rb2	7.20 cc	Yes
			Rb3	4.37 cc	Yes
	Volume > 27.5 Gy	≤ 0.03 cc	Rb1	0.00 cc	Yes
			Rb2	0.00 cc	Yes
			Rb3	0.02 cc	Yes
**Brain**	Volume = 10 Gy	< 10 cc	Rb1	0.00 cc	Yes
			Rb2	1.50 cc	Yes
			Rb3	2.02 cc	Yes

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
