# Peer review of "Development and Validation of a Clinically Relevant Workflow for MR-Guided Volumetric Arc Therapy in a Rabbit Model of Head and Neck Cancer"

_cancers, 2020, doi:10.3390/cancers12030572_

Round 1

Reviewer 1 Report

I thank both the Editor and authors for this manuscript.

Overall :

I agree with that MRgRT could become a game changer for some diseases and techniques such as SRS.

Here, authors have well develloped a new workflow with MRI for SRS H&N cancer.

However, 2D RT without CT delineation was used for comparison but authors should rather used VMAT and CT delineation instead.

Abstract : CE (figure for contrast enhanced) in CE-MRI is not described. No method is described for comparison of delineation "enabled improve tumor targeting ability and minimal dose to organs at risk" explaining L32 results.

Introduction : 2D-RT is not the standard for HNSCC RT in human but should be described here as the standard for large animals.

Methods :

Is CT enhanced ?

CE-MRI is not described as dedicated to RT delineation and could present some distorsions.

"As evident from the individual images, CE-MRI provided excellent soft tissue contrast compared to CT" should be provent by inter- and intra- examinator tests.

Registration between MRI and CT could result in bias. Is rigid deformation applied ?

Authors should used VMAT and CT delineation to compared both for DVH and outcome if they want to show the impact of MRgRT.

Results : Rb1,2 or 3 are not described...

Therapeutic benefit of MR-VMAT should not be evaluated because survival was different between the different groups. In addition, since D95% seems to be the same for both group, results should be the same for tumor control.

Discussion: Nothing

Conclusion : Authors should not extrapolated results to fractionnated regimen as they have only experimented SRS. Moreover because 2D RT is not the standard of care for human nothing should be said about MRgRT impact, CT based VMAT may lead to same results...Nothing could be said on delineation.

Author Response

  1. Overall: I agree with that MRgRT could become a game changer for some diseases and techniques such as SRS. Here, authors have well developed a new workflow with MRI for SRS H&N cancer. However, 2D RT without CT delineation was used for comparison but authors should rather used VMAT and CT delineation instead. We appreciate the reviewer’s comment. Our rationale for comparing MRgRT to 2D RT was to illustrate the benefit of our technique compared to the conventional orthovoltage systems used for radiation delivery in most preclinical studies. These orthovoltage systems do not employ image guidance. Nevertheless, the reviewer’s point is well taken. We have therefore included DVH for CT based plans and described our comparative assessment of the CT and MR based plans (Supplementary Fig. A4).
  1. Abstract: CE (figure for contrast enhanced) in CE-MRI is not described. No method is described for comparison of delineation "enabled improve tumor targeting ability and minimal dose to organs at risk" explaining L32 results. The sentence in reference to CE-MRI in the Abstract has been revised accordingly.
  1. Introduction: 2D-RT is not the standard for HNSCC RT in human but should be described here as the standard for large animals. We have now added a line in the introduction (Lines 56-58) to explicitly state this point.
  1. Methods: Is CT enhanced? CE-MRI is not described as dedicated to RT delineation and could present some distortions. The reviewer’s comment regarding potential distortion is well taken. In our study, CE-MR images were rigidly co-registered to non-contrast enhanced CT images. While image registration was carefully performed to ensure that pixel-to-pixel relationships were maintained, the possibility of some distortion cannot be ruled out. We have now recognized this in the ‘Discussion’ section of the revised manuscript (Lines 224-227).
  1. As evident from the individual images, CE-MRI provided excellent soft tissue contrast compared to CT" should be proven by inter- and intra- examiner tests. We appreciate the reviewer’s suggestion. While it would have been valuable to compare inter- and intra-examination, the contouring and treatment plans were performed by one medical physicist (E.R.B). We have therefore modified the sentence.
  1. Registration between MRI and CT could result in bias. Is rigid deformation applied? The reviewer is correct. Rigid deformation was applied to co-register the MR and CT images. As stated in our response to critique #4, potential distortions cannot be completely ruled out. However, the short-time interval between the imaging and the RT delivery (24h or less) ensured minimal change in tumor volume lowering the bias or risk of errors.
  1. Authors should used VMAT and CT delineation to compared both for DVH and outcome if they want to show the impact of MRgRT. As suggested by the reviewer, we have now performed delineation and treatment planning using CT and compared DVH between CT and MRI guided planning. This result has now been included in Supplementary Fig A4 (Lines 141-145).
  1. Results: Rb1,2 or 3 are not described...We regret this error. We have now described the individual rabbit IDs (Page 5, line 151) presented in the tables.
  1. Therapeutic benefit of MR-VMAT should not be evaluated because survival was different between the different groups. In addition, since D95% seems to be the same for both group, results should be the same for tumor control. Thank you for pointing this out. This is a fair comment. We have therefore removed statements referring to the therapeutic impact. 
  1. Conclusion : Authors should not extrapolate results to fractionated regimen as they have only experimented SRS. Moreover because 2D RT is not the standard of care for human nothing should be said about MRgRT impact, CT based VMAT may lead to same results...Nothing could be said on delineation. This is also fair critique. Recognizing the validity of the critique, we have now removed statements regarding impact of MRgRT.

Reviewer 2 Report

Dear authors,

Thank you for allowing me to review your manuscript.  You have brought to my attention the relevant current day issues regarding animal model testing for radiation therapy.  As clinical radiotherapy advances with image guidance, VMAT delivery, and the incorporation of sophisticated imaging for planning guidance, it is of utmost importance that pre-clinical models evaluate radiotherapy planning and delivery with the same sophisticated techniques available, as this study has shown.

Please consider the following points for editing and clarification:

Figure 1 legend lines 83-86 please spell out MRI, CT, and US and symbol on d1 assuming that is radiation dose delivery for ease to reader

Table 1 % PTV limit please indicate if >/= 95; conformality limit please indicate </= 1.2; max dose beyond PTV limit please indicate </=; vol 50%/vol PTV limit please indicate <, also was actual really 1 in all 3 rabbits as that is hard to believe

Table 2 on limits please place < for spinal cord, trachea, skin; please also define how skin OAR was defined, was it surface – 2mm or surface alone?

Figure 5 please indicate column heading d = day and in legend utilize days not weeks for ease of interpretation

Author Response

Thank you for allowing me to review your manuscript.  You have brought to my attention the relevant current day issues regarding animal model testing for radiation therapy.  As clinical radiotherapy advances with image guidance, VMAT delivery, and the incorporation of sophisticated imaging for planning guidance, it is of utmost importance that pre-clinical models evaluate radiotherapy planning and delivery with the same sophisticated techniques available, as this study has shown. We appreciate the positive and valuable feedback provided by the reviewer.

  1. Figure 1 legend lines 83-86 please spell out MRI, CT, and US and symbol on d1 assuming that is radiation dose delivery for ease to reader. Thank you. We have revised the figure legend based as suggested by the reviewer.
  1. Table 1 % PTV limit please indicate if >/= 95; conformality limit please indicate </= 1.2; max dose beyond PTV limit please indicate </=; vol 50%/vol PTV limit please indicate <, also was actual really 1 in all 3 rabbits as that is hard to believe. We regret these errors. The corrections have been made to Table 1. The number for vol 50%/vol PTV was calculated by dividing the 15Gy vol (cc) to PTV volume (cc) and was 1 for all three rabbits.
  1. Table 2 on limits please place < for spinal cord, trachea, skin; please also define how skin OAR was defined, was it surface – 2mm or surface alone?Thank you. Table 2 has been revised accordingly. The reviewer is correct. Skin was defined as surface-2mm and has now been defined in the table.
  1. Figure 5 please indicate column heading d = day and in legend utilize days not weeks for ease of interpretation. Thank you for your comment. We have revised the figure legend accordingly.